# OrbitGrasp: SE(3)-Equivariant Grasp Learning

**Boce Hu[1], Xupeng Zhu[⋆1], Dian Wang[⋆1], Zihao Dong[⋆1], Haojie Huang[⋆1],**
**Chenghao Wang[⋆1], Robin Walters[1,2], Robert Platt[1,2]**
[1]Northeastern University    [2]Boston Dynamics AI Institute
https://orbitgrasp.github.io/

**Abstract:** While grasp detection is an important part of any robotic manipulation pipeline, reliable and accurate grasp detection in SE(3) remains a research challenge. Many robotics applications in unstructured environments such as the home or warehouse would benefit a lot from better grasp performance. This paper proposes a novel framework for detecting SE(3) grasp poses based on point cloud input. Our main contribution is to propose an SE(3)-equivariant model that maps each point in the cloud to a continuous grasp quality function over the 2-sphere $S^2$ using a spherical harmonic basis. Compared with reasoning about a finite set of samples, this formulation improves the accuracy and efficiency of our model when a large number of samples would otherwise be needed. In order to accomplish this, we propose a novel variation on EquiFormerV2 that leverages a UNet-style encoder-decoder architecture to enlarge the number of points the model can handle. Our resulting method, which we name *OrbitGrasp*, significantly outperforms baselines in both simulation and physical experiments.

**Keywords:** Grasp Detection, Equivariance, Symmetry, Grasp Learning

## 1 Introduction

The ability to detect and evaluate good grasp poses in a manipulation scene is a critical part of robotic manipulation. Despite extensive recent work in the area, e.g. [1, 2, 3, 4, 5, 6, 7, 8, 9], grasp detection is still not accurate and reliable enough for many practical applications. A key challenge here is effective reasoning over hand poses in SO(3), i.e. the three dimensions of orientation spanned by SE(3). Simply representing orientations in SO(3) can be a challenge due to classic problems like gimble lock [10] and discontinu-

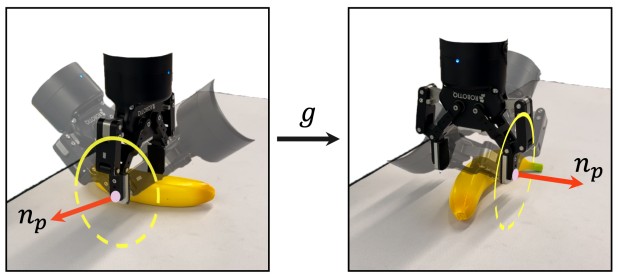

Figure 1. We infer an orbit of grasps (yellow ellipse) defined relative to the surface normal (red arrow) at the contact point (pink dot). Since our model is equivariant over SO(3), the optimal pose (represented by the solid gripper) on the orbit rotates consistently with the scene (left and right show a rotation by 90 degrees).

ity [11]. More importantly, it would be very helpful to be able to infer grasp quality over a continuous range of orientations in SO(3) – something which is a major challenge for conventional grasp methods. Finally, since SE(3) grasping problems are often equivariant (i.e. symmetric) in SO(3), it is often desirable to encode symmetry assumptions into the model.

This paper addresses these challenges by leveraging recent innovations in equivariant point models [12, 13, 14, 15]. Specifically, we use a model that maps each point in a point cloud to a grasp

---

⋆ Equal contribution.

8th Conference on Robot Learning (CoRL 2024), Munich, Germany.

quality function over the 2-sphere $S^2$. For each point in the cloud, this function encodes grasp quality for the space of possible hand approach directions toward that point. An important question here is how to represent this quality function over $S^2$. Following recent work [1, 16, 8], our approach is to leverage the spherical harmonic basis. Specifically, at each point in the cloud, the neural network outputs a vector of Fourier coefficients to the spherical harmonic basis functions, thereby defining a grasp quality function over hand approach directions. With this learned per-point grasp distribution, we can evaluate the quality of a large number of potential grasp poses quickly and easily, thereby more easily locating a grasp appropriate for a given downstream manipulation task.

To summarize, this paper introduces several key contributions. First, we propose a novel method of using spherical harmonics to reason about an *orbit* (i.e. a $S^1$ manifold embedded in $S^2$) of grasp approach directions defined relative to the surface normal at each contact point in the point cloud, as shown in Figure 1. Second, we enhance the recently developed SE(3)-equivariant model, EquiFormerV2 [17], by incorporating a U-Net backbone and thereby enabling the model to accommodate a larger number of points in the point cloud. Finally, we evaluate our method against multiple established baselines [5, 18, 8, 9], using the benchmark tasks proposed in [5]. The results indicate that our approach, which we name *OrbitGrasp*, convincingly outperforms the baselines in both single-view and multi-view settings, in both simulation and physical experiments.

## 2 Related Work

SE(3) **Grasp Detection:** Current 6-DoF grasping tasks in cluttered tabletop scenarios primarily use a volumetric-based or point cloud-based scene representation as input and output one or more optimal grasp poses [7, 3, 19, 2]. Volumetric Grasping Network (VGN) [5] and Grasp detection via Implicit Geometry and Affordance ( GIGA) [18] use 3D convolutional models to reason about a 3D truncated signed distance function (TSDF). However, these methods suffer from high memory consumption and resolution limitations. In contrast, point clouds can provide higher resolution but are more difficult to reason about due to the lack of structure. One approach to adding structure is to leverage equivariant point cloud models [20, 21]. For example, EdgeGrasp [8] introduced a method based on a vector neuron model. CAPGrasp [22] is another equivariant model that samples grasp candidates while constraining the gripper approach direction to be within a certain angular distance from the surface normal at the associated point. Other recent grasping methods leverage surface reconstruction of the object to be grasped, including NeuGraspNet [23] and ICGNet [9].

**Equivariance in Robot Learning:** Using symmetry, i.e. equivariance, in robot learning tasks has recently been used to improve sample efficiency and generalization. [24, 25, 26] introduce equivariant neural network models in the form of steerable convolutional layers [27, 28] to SE(2) manipulation tasks. [29] applies similar models to transporter network [30], making it bi-equivariant and significantly improving sample efficiency. EDF [31] introduces an equivariant energy model that produces a continuous distribution over pose. Fourtran [32] extends [29] from SE(2) to SE(3) using a Fourier representation of rotations. [33] develops a novel, dense, interpretable representation for relative object placement tasks. RiEMann [34] presents the first near real-time SE(3)-equivariant robot manipulation framework for point cloud inputs that addresses the problem of slow inference over the SE(3)-manifold. Most relevant to this paper are [1] who obtain SE(2) equivariance using steerable convolutional layers and [8] who use vector neurons to obtain SE(3) equivariance.

**OrbitGrasp:** The method proposed in this paper, *OrbitGrasp*, is distinct from the work described above in a couple of important ways. First, prior work in grasp detection is generally sample-based [35, 7, 6], i.e., they evaluate grasp quality for a (large) finite set of samples or discrete voxel/pixel locations. In contrast, OrbitGrasp evaluates grasp quality for a *continuous* range of approach directions by inferring the parameters of spherical harmonic basis functions over $S^2$. This is simultaneously more computationally efficient and more precise than sample-based methods. Another distinction between OrbitGrasp and prior work is that the representation of rotation in terms of high degrees of spherical harmonics encodes continuous equivariant constraints more accurately and, therefore, models the grasp function better.

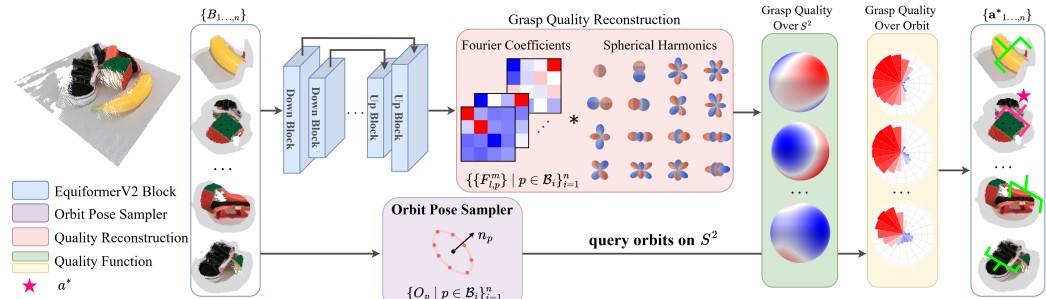

Figure 2. OrbitGrasp takes the point cloud $B_i$ (a neighborhood around center point $c_i$) as input and outputs a grasp quality function $f_p \colon S^2 \to \mathbb{R}$ for each point $p$ in $B_i$. The model produces Fourier coefficients for each $p$ (represented as different channels in the network output), which are used to reconstruct $f_p$ based on spherical harmonics, as in Equation 1. The Orbit Pose Sampler generates multiple poses for each $p$ perpendicular to the surface normal $n_p$ and queries corresponding $f_p(\cdot)$ to evaluate these grasp qualities along the orbit. The grasp with the highest quality is then selected, thereby producing the optimal grasp pose $a^*$, as shown on the right.

## 3 Background

We introduce concepts from group theory and SE(3)-equivariant neural networks, which are key to our method. For additional background, see, e.g., [36].

**Representations of** SE(3)**:** Our work focuses on the special Euclidean group $\mathrm{SE}(3) = \mathrm{SO}(3) \ltimes \mathbb{R}^3$ of 3D rotations and translations. A representation of a group $G$ is a homomorphism $\rho \colon G \to \mathrm{GL}(V)$ mapping each group element $g$ to an invertible linear operator on the vector space $V$. The representation $(\rho, V)$ is an irreducible representation, or irrep, if no non-trivial subspace $W \subset V$ is closed under the action of $G$. The irreps of $\mathrm{SO}(3)$ are classified by positive integers $l \in \mathbb{Z}_{\geq 0}$ with the irrep $V_l$ of type $l$ having dimension $2l+1$. For example, $l = 1$ gives the standard rotation action on $\mathbb{R}^l$. The group action on $V_l$ is given by the Wigner D-matrices $D^l \colon \mathrm{SO}(3) \to \mathrm{GL}(V_l)$ of size $(2l+1) \times (2l+1)$. Traditionally, the Wigner D-matrices are parameterized by Euler angles $(\alpha, \beta, \gamma)$ and indexed symmetrically about 0 as $D^l_{m'm}(\alpha, \beta, \gamma)$ where $-l \leq m, m' \leq l$.

**Spherical Harmonics:** The real spherical harmonics (SH) are functions on the sphere $Y_l^m \colon S^2 \to \mathbb{R}$ indexed by $l \in \mathbb{Z}_{\geq 0}$ and $-l \leq m \leq l$. As a set, they define an orthonormal basis of the Hilbert space of square-integrable functions on the sphere $L^2_{\mathbb{R}}(S^2) = \{f \colon S^2 \to \mathbb{R} : \int_{S^2} |f|^2 < \infty\}$. That is, given $f \colon S^2 \to \mathbb{R}$, we can write $f(\theta, \phi) = \sum_{l=0}^{\infty} \sum_{m=-l}^{l} \mathcal{F}_l^m Y_l^m(\theta, \phi)$ where $\mathcal{F}_l^m$ are the Fourier coefficients of $f$. For a finite truncation at $l \leq L$, the total number of Fourier coefficients is $(L+1)^2$. The mapping $\mathrm{FT} \colon f \mapsto \{\mathcal{F}_l^m\}$ is known as the spherical Fourier transform. Approximating $f$ by storing the $\mathcal{F}_l^m$ up to degree $l \leq L$ gives an efficient way to encode the spherical signal $f$. The spherical harmonics are compatible with the $\mathrm{SO}(3)$ action on the sphere, making them suitable for use in $\mathrm{SO}(3)$-equivariant networks. The rotation of a spherical signal $f$ can be computed in Fourier space using Wigner D-matrices. Let $g \in \mathrm{SO}(3)$. Define $f' = g \cdot f$ to be the rotated spherical signal defined $(g \cdot f)(u) = f(g^{-1} \cdot u)$ for $u \in S^2$. Denote the Fourier coefficients of $f'$ by $\mathrm{FT}(f') = \{\mathcal{F}_l^{m'}\}$. Then in terms of the Fourier coefficients of $f$ we have $\mathcal{F}_l' = D^l(g) \mathcal{F}_l$ where $\mathcal{F}_l$ denotes the vector $\mathcal{F}_l = (\mathcal{F}_l^m)_{m=-l}^l$.

**EquiFormerV2:** EquiFormerV2 [17] is a SE(3)-Equivariant Graph Neural Network (GNN) that addresses the high computational costs and poor generalization of higher-degree irreps in EquiFormer [37]. Unfortunately, since EquiFormer and EquiFormerV2 were developed for applications in computational biology, they have a hard time scaling to point cloud data, which generally contain more points than molecules do atoms. To address this limitation, our paper proposes a variation of EquiFormerV2 that uses a U-Net architecture to increase the capacity of the model. Compared to [38], our new model can reason about higher-degree spherical harmonic functions.

## 4 Method

**Problem Statement:** Given a point cloud $\mathcal{P} \subset \mathbb{R}^3$ captured by one or multiple depth cameras, our objective is to identify a set of good grasp poses. Specifically, we want to estimate a function

$\Gamma\colon (\mathcal{P}, a) \mapsto [0, 1]$ which maps the point cloud $\mathcal{P}$ and the hand pose $a \in \mathrm{SE}(3)$ to the probability of successful grasp. Notice that we expect $\Gamma$ to be invariant to translations and rotations $g \in \mathrm{SE}(3)$, i.e. we expect $\Gamma(g \cdot \mathcal{P}, g \cdot a) = \Gamma(\mathcal{P}, a)$. This reflects an assumption that the probability that a grasp is successful does not change when both the scene and the grasp pose transform in concert.

**Summary of Approach:** Figure 2 illustrates our model and approach. First, we downsample and crop the point cloud into a small number of point neighborhoods $B_1, \ldots, B_k$. Then, we evaluate our model for each $B_i$. For each point $p \in B_i$, the model predicts the grasp quality over the space of possible hand approach directions, represented as a function $f_p\colon S^2 \to \mathbb{R}$. We densely sample this function over an *orbit* of approach directions orthogonal to the object surface normal at $p$ and obtain a single grasp that maximizes quality over the orbit. The result is a single-hand orientation in $\mathrm{SO}(3)$ at each point in the cloud, corresponding to a good grasp at that point. This produces a dense sampling of grasps that can be filtered further for grasping or other downstream tasks. Figure 7 in the Appendix provides a detailed, zoomed-in visualization. This approach has several important advantages. First, since our model outputs a *continuous* distribution over $S^2$, it generalizes better over orientation than would a sample-based approach. Second, since our model is $\mathrm{SE}(3)$-equivariant, it incorporates problem symmetry as an inductive bias into the model. Finally, the approach only considers grasps that make contact parallel to the object surface normal, thereby incorporating an additional geometric prior.

**Sampling $k$ Center Points:** We prepare the point cloud for processing by the prediction model as follows. First, we downsample the input point cloud $\mathcal{P}$ to a tractable number of points $\bar{\mathcal{P}}$ (around 4k to 6k). Rather than passing the entire point cloud through our model (which would be expensive), we define smaller point clouds $B_1, \ldots, B_k \subset \bar{\mathcal{P}}$ which we evaluate over separately ($k$ is usually around 10). The $B_i$ are defined as neighborhoods of $k$ center points $c_1, \ldots, c_k$ from $\bar{\mathcal{P}}$ selected as follows. If a segmentation of the point cloud into objects is available, then these $k$ center points could be the 3D positions of the centers of individual object masks. Otherwise, these centers could be random samples obtained using farthest point sampling (FPS). In our experiments, we generate center points using object masks during training to enhance the model's object-centric awareness, and FPS during inference to demonstrate generalization and performance without segmentation. A detailed description of our mask-based sample generation is provided in Appendix A. We then construct each $B_i$ as a neighborhood around $c_i$ as follows. First denote the set of points contained within radius $r_l$ ball centered at $c_i$ by $\mathcal{B}_i = \mathcal{B}(c_i, r_l) = \{p \in \bar{\mathcal{P}} \mid \|p - c_i\| \leq r_l\}$ for a parameterized radius $r_l$. Since the grasp quality at a point $p$ depends on the geometry of a sufficiently large neighborhood of $p$ to avoid boundary effect, we then set $B_i = \mathcal{N}(c_i, m)$ to be the $m$ nearest neighbors of $c_i$, where $m$ is chosen large enough such that $\mathcal{B}_i \subset B_i$ and $B_i$ is larger by a fair margin. The larger point cloud $B_i$ is passed as input to the model, but the output grasp quality is only inferred over the smaller point cloud $\mathcal{B}_i$. This assures inference is only performed on points with sufficient geometric context.

**Representing Grasp Quality Over $S^2$:** We model the grasp quality function $\Gamma\colon (\mathcal{P}, a) \mapsto [0, 1]$ with a neural network $\bar{\Gamma}\colon B_i \mapsto \{f_p\colon S^2 \to \mathbb{R} \mid p \in B_i\}$ that takes each of the $k$ neighborhoods $B_1, \ldots, B_k$ in $\bar{\mathcal{P}}$ as input. For each $B_i$, the model outputs a spherical function $f_p\colon S^2 \to \mathbb{R}$ for each $p \in B_i$. The function $f_p$ represents the grasp quality over all approach directions in $S^2$ at the point $p$. The function $f_p$ is represented by Fourier coefficients of the spherical harmonics,

$$f_p(\theta, \phi) = \sum_{l=0}^{n} \sum_{m=-l}^{l} \mathcal{F}_{l,p}^m Y_l^m(\theta, \phi). \tag{1}$$

where $Y_l^m(\theta, \phi)$ denotes the spherical harmonic of degree $l$ and order $m$ evaluated at the point on $S^2$ defined in spherical coordinates $(\theta, \phi)$. Concretely, the model outputs the set of Fourier coefficients $\{\mathcal{F}_{l,p}^m\}$ for at each $p \in B_i$.

**Implementing $\bar{\Gamma}$ as an Equivariant Neural Network:** We implement $\bar{\Gamma}$ using a modified version of EquiFormerV2 [17], a fully $\mathrm{SE}(3)$-equivariant, GNN-based network designed for handling node-based data. In the original EquiFormerV2 [17], all nodes are fully connected. However, in order to handle the larger number of points in our point clouds, we developed a UNet-style version of the model, which has a greater capacity (inspired by [38]). Figure 2 illustrates the model architec-

ture. Given the point cloud $B_i$, successive blocks in our model sequentially downsample the points using FPS. As the number of points decreases, the radius of the edge in the graph increases, allowing distant points to communicate, expanding the receptive field per block, and capturing global features. To recover the features of each point in the original point cloud after downsampling, we upsample point features back by inverting the edges: during downsampling, we record the source and target edges in each block, and during upsampling, we swap these edges. This ensures that features gathered during downsampling are effectively propagated back to the original points. In addition, we employ skip connections between each downsample block and its corresponding upsample block to aggregate features and prevent degradation. Details of the architecture are in Appendix B. This model satisfies the desired SE(3) equivariance constraint described in Section 4, $\bar{\Gamma}(gB_i) = g\bar{\Gamma}(B_i) = \{g \cdot f_p \colon S^2 \to \mathbb{R} \mid p \in gB_i\}$. For $g \in \mathrm{SO}(3)$, if the input point cloud is rotated $gB_i$, the equivariance constraints ensure the output Fourier coefficients are also transformed $D^l(g)\mathcal{F}_{l,p}$. Evaluating at a transformed grasp $g \cdot u$ thus gives the same quality of a successful grasp $(g \cdot f_p)(g \cdot u) = f_p(g^{-1} \cdot g \cdot u) = f_p(u)$.

**Inferring Grasp Pose:** Since our neural network model only infers grasp quality over $S^2$, we must obtain the remaining orientation DoF somehow. Following [8], we accomplish this by constraining one of the two gripper fingers to make contact such that the object surface normal at the contact point is parallel to the gripper closing direction (see Figure 3). Specifically, for a point $p \in \mathcal{B}_i \subset B_i$ in region $B_i$ with object surface normal $n_p$, we constrain the hand y-axis (gripper closing direction) to be parallel to $n_p$. Therefore, valid hand orientations form a submanifold in $\mathrm{SO}(3)$ homeomorphic to a 1-sphere $S^1$ which we call the *orbit* at $p$

$$O_p = \{R = [r_1, n_p, r_3] \in \mathrm{SO}(3)\} \qquad (2)$$

where $r_1, n_p, r_3$ are the columns of the 3-by-3 rotation matrix $R$. Valid orientations are determined by the $z$-axis of the gripper (the

Figure 3. Green and blue denote the $y$, $z$ directions of the hand, and $n_p$ is the normal vector at $p$ (red). Black is the orbit of the approach direction.

approach direction of the hand) which may be any unit vector perpendicular to $n_p$. We may thus specify valid grasps by their approach vector $r_3 \in \overline{O}_p = \{r_3 \in S^1 : n_p^\top r_3 = 0\}$ since $r_1 = -n_p \times r_3$. The details of how we evaluate the $f_p$ can be found in Appendix C.

In the above discussion, notice that we have constrained *two* DoFs of orientation, which could suggest that our model $f_p$ should infer only one DoF of orientation, not two. This raises the question: Why is $f_p$ defined over $S^2$ (the spherical harmonics) rather than $S^1$ (the circular harmonics)? The answer is interesting and important. Defining the output of $f_p$ over $S^1$ would require choosing a *gauge*, i.e. a mapping of $S^1$ to the tangent plane at each point. Assuming $S^1$ were parameterized by an angle $\theta$, we would need to select a vector that defined $\theta = 0$. Notice that it would be hard to make this selection in a consistent way across the point cloud. In contrast, by defining the output of $f_p$ over $S^2$, we avoid this problem because we can express this function in a single global coordinate system for all points in the cloud. One way to think about this is that we have gained consistency in representation by adding an extra dimension, i.e. going from $S^1$ to $S^2$.

## 5   Experiments

**Simulation Environment:** We evaluated our model in simulation using PyBullet [39]. Our setting is the same as that used in [18, 5, 8, 9]. The workspace size is a 30 $cm^3$ cube and contains varying numbers of objects randomly placed within it. The object dataset includes 303 training and 40 test objects from [40, 41, 42, 43]. A Franka-Emika Panda floating gripper is used to grasp objects in the workspace. We evaluated two camera configurations. The first is a single-view setting where a camera is positioned randomly on a spherical region around the workspace. The second is a three-camera multi-view setting. Following [5], we evaluate two different grasping tasks: a **_Pile_** setting where objects are randomly dropped into the workspace and a **_Packed_** setting where objects are placed upright in random poses. For more details, see Appendix D.

| Setting | Method | Packed | | Pile | |
|---|---|---|---|---|---|
| | | GSR (%) | DR (%) | GSR (%) | DR (%) |
| Single-view | GIGA [18] | $89.9 \pm 1.7$ | $87.6 \pm 2.0$ | $76.3 \pm 2.4$ | $80.9 \pm 4.1$ |
| | GIGA-HR [18] | $91.4 \pm 1.5$ | $88.5 \pm 1.4$ | $86.5 \pm 1.2$ | $80.8 \pm 1.9$ |
| | EdgeGrasp [8] | $92.5 \pm 0.9$ | $94.3 \pm 1.1$ | $91.5 \pm 1.3$ | $92.5 \pm 1.3$ |
| | VNEdgeGrasp [8] | $91.6 \pm 1.7$ | $94.4 \pm 1.5$ | $92.0 \pm 1.8$ | $92.2 \pm 2.1$ |
| | ICGNet [9] | $97.7 \pm 0.9$ | $97.5 \pm 0.3$ | $\underline{92.0 \pm 2.6}$ | $94.1 \pm 1.4$ |
| | OrbitGrasp (3M) | $\mathbf{98.4 \pm 0.5}$ | $\mathbf{98.8 \pm 0.3}$ | $\mathbf{96.3 \pm 0.3}$ | $\underline{97.7 \pm 0.7}$ |
| | OrbitGrasp (6M) | $\underline{98.3 \pm 0.7}$ | $\mathbf{98.8 \pm 0.6}$ | $\mathbf{96.7 \pm 1.1}$ | $\mathbf{97.9 \pm 0.5}$ |
| Multi-view | VGN [5] | $89.8 \pm 2.0$ | $82.6 \pm 3.2$ | $63.2 \pm 1.1$ | $45.6 \pm 0.5$ |
| | VNEdgeGrasp [8] | $97.1 \pm 1.3$ | $96.1 \pm 0.5$ | $95.1 \pm 1.0$ | $95.5 \pm 1.5$ |
| | OrbitGrasp (3M) | $\underline{98.6 \pm 0.3}$ | $\underline{99.1 \pm 0.5}$ | $\mathbf{98.6 \pm 0.7}$ | $\mathbf{98.5 \pm 0.5}$ |
| | OrbitGrasp(6M) | $\mathbf{99.0 \pm 0.6}$ | $\mathbf{99.2 \pm 0.3}$ | $\underline{98.5 \pm 0.6}$ | $\underline{98.2 \pm 0.6}$ |

Table 1. We compared the OrbitGrasp with various baselines in terms of grasp success rate (GSR) and declutter rate (DR), the same metrics as used in [8]. For the single-view setting, we tested pretrained models from [8] and obtained results for GIGA, GIGA-HR, and ICGNet directly from [9]. For the multi-view setting, we retrained VNEdgeGrasp and tested the pretrained VGN model for comparison. OrbitGrasp (3M) is trained on the 3M dataset, which is similar in size to that used to train ICGNet and EdgeGrasp. OrbitGrasp (6M) is trained on the full 6M dataset and reaches a slightly higher level of performance. The best results are marked with **bold** and the second best results are underlined.

**Generating Training Data:** We generate data in simulation by loading a random number of objects into the workspace and then generating point clouds in both single-view (one depth camera) and multi-view settings (three depth cameras spaced evenly around the workspace). We add Gaussian noise sampled from $N(0, 0.001)$ to the point cloud to make the model more robust to real world sensor noise. After creating the scenes, we must generate training data for the grasp model. First, object masks are obtained using the Segment Anything Model (SAM) [44] (see Appendix A). We use SAM to generate object masks for training only, and for inference, SAM is not required. For each masked object, we randomly select a set of candidate grasp points and evaluate 36 candidate grasp orientations per point that satisfy Equation 2, i.e. where the gripper $z$ axis is orthogonal to the object surface normal at contact. In total, approximately 6M grasp poses (approximately 2.5M positive and 3.5M negative) are generated for each camera setting: 4M from 2,500 *Pile* scenes and 2M from 800 *Packed* scenes. The data is split with 90% for training and 10% for validation.

**Training Details:** Since our training data is segmented into objects, we use the center of every mask as the center point $c_i$ and obtain the local point neighborhood $B_i$. During training, we balance the positive and negative grasp labels in each $B_i$. The model is trained using the AdamW optimizer[45], starting with a learning rate of 1e-4 and using a cosine annealing scheduler [46]. We apply binary cross-entropy loss for each grasp pose and a dropout rate [47] of 0.1 to prevent overfitting. Our network is trained for 15 epochs, taking approximately 25 hours on 22k point clouds. Each SGD step takes 0.25 seconds with a batch size of 1. The model is trained on an NVIDIA RTX 4090 GPU.

## 5.1 Comparison With Baseline Methods in Simulation

**Baselines:** We compare our method with several strong baselines. In the single random view setting, we compare it with volumetric-based methods: GIGA and GIGA-HR (high resolution) [18]. Since GIGA was trained under a fixed view, we compare it with the results from [9], which retrains it under a random view setting. We also compare our method with EdgeGrasp and VNEdgeGrasp [8], two point cloud-based methods, where VNEdgeGrasp is SE(3)-invariant. Finally, we compare our method with ICGNet [9], an approach focused on partial observation to object-centric grasping that reconstructs the full 3D shape of objects before finding grasps. In the multi-view setting, we compare our method with VGN [5], which takes a TSDF as input and generates a single grasp pose per voxel, as well as VNEdgeGrasp [8] trained in the multi-view setting.

**Results:** We report the comparison result in Table 1 under two different settings, single-view and multi-view, and two different tasks, *Packed* and *Pile*. For a fair comparison with the baselines, we also trained our model on a similar amount of data (3M labeled grasp poses uniformly randomly downsampled from the original 6M dataset.), consistent with EdgeGrasp [8] and ICGNet [9]. Additional performance details under different amounts of data are pro-

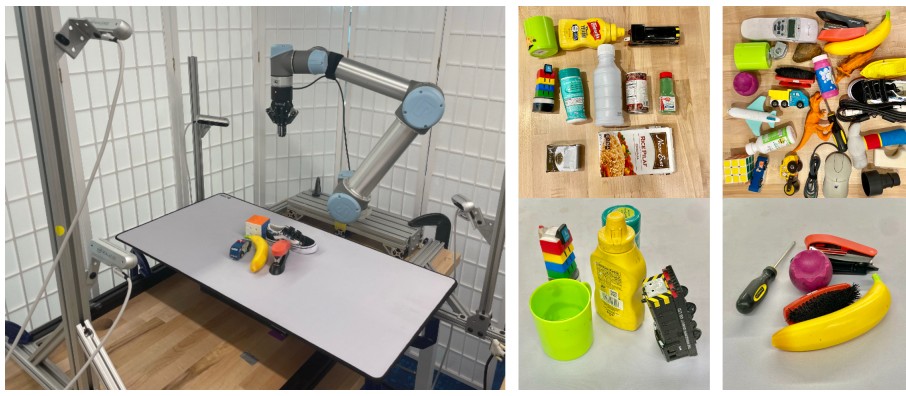

| (a) | (b) | (c) |

Figure 4. **Real world Experiment Setting.** (a) Robot platform setup. (b) Upper: Packed object set with 10 objects. Bottom: Packed scene (c) Upper: Pile object set with 25 objects. Bottom: Pile scene

vided in the Appendix E. Performance is evaluated based on two metrics: (1) Grasp Success Rate (GSR) = num of successful grasps/num of total grasps, and (2) Declutter Rate (DR) = num of grasped objects/num of total objects. For each task, we conducted five iterations of testing, with each iteration containing 100 rounds (5 objects per round). A round ends when either all objects are cleared or two consecutive grasp failures occur.

The results in Table 1 indicate that our method outperforms all baselines across both settings and tasks in terms of both GSR and DR, in both the 3M and 6M training sets. The high GSR indicates that our model can predict accurate grasp quality. On the other hand, the high DR signifies that our model infers accurate grasp poses that do not move objects outside of the workspace. Notably, our model performs well even though point centers are selected differently at training time (centered on object segments) and test time (selected using FPS), i.e., as described in Section 4. This demonstrates that the grasp quality function $\bar{\Gamma}$ output by our method exhibits strong robustness to different point cloud geometries.

## 5.2 Physical Experiments

| Setting | Method | Packed | | Pile | |
| --- | --- | --- | --- | --- | --- |
| | | GSR (%) | DR (%) | GSR (%) | DR (%) |
| Single-view | VNEdgeGrasp [8] | 88.9(96/108) | 96.0(96/100) | 88.2(97/110) | 97.0(97/100) |
| | OrbitGrasp | **92.5(98/106)** | **98.0(98/100)** | **93.2(97/104)** | **98.0(98/100)** |
| Multi-view | VNEdgeGrasp [8] | 90.1(100/111) | **100.0(100/100)** | 92.1(93/101) | 97.0(97/100) |
| | OrbitGrasp | **95.2(100/105)** | **100.0(100/100)** | **94.3(99/105)** | **99.0(99/100)** |

Table 2. We compared the results of OrbitGrasp with VNEdgeGrasp [8] using the same metrics as in the simulation experiments. Our experiment settings for single-view and multi-view are shown in Figure 4. Sometimes, a single trial resulted in grasping two objects simultaneously.

To assess our method's real-world performance, we conduct physical experiments involving two tasks under two camera settings, replicating those in the simulation. We directly transfer the trained model from the simulation to the real-world setting to evaluate the performance gap between them.

**Setup:** As shown in Figure 4 (a), a UR5 robot arm is equipped with a Robotiq-85 Gripper. We mount and calibrate 4 RealSense D455 cameras on the table. In the multi-view setting, 3 low-mounted cameras capture the scene for full observation. In the single view setting, we use the top-mounted camera. After obtaining the point cloud, we crop it to the workspace, filter outliers, and calculate the surface normals at each point. We do *not* crop the table from our point clouds. Objects are placed on the table as follows. In the ***Packed*** setting, we draw 5 objects uniformly at random from the set of 10 objects shown at the top of Figure 4 (b) and place them in upright orientations with a randomly selected planar position and orientation. In the ***Pile*** setting, we draw 5 objects uniformly at random from the set of 25 objects shown at the top of Figure 4 (c) and dump them onto a tabletop out of a box. We conduct 10 rounds of experiments for the ***Packed*** scene and 4 rounds for the ***Pile*** scene for each view setting. We configure our method as follows. We use $k = 10$ center points selected

| Model | #Params | Packed | | Pile | |
|---|---|---|---|---|---|
| | | GSR (%) | DR (%) | GSR (%) | DR (%) |
| PointNet++ [48] | 14M | $78.1 \pm 2.9$ | $78.8 \pm 1.9$ | $60.5 \pm 1.8$ | $54.1 \pm 2.7$ |
| OrbitGrasp ($l = 1, m = 1$) | 4M | $94.2 \pm 1.2$ | $96.3 \pm 1.3$ | $93.9 \pm 0.5$ | $97.0 \pm 0.7$ |
| OrbitGrasp ($l = 2, m = 2$) | 8M | $97.4 \pm 0.5$ | $98.5 \pm 0.6$ | $96.3 \pm 1.0$ | $97.6 \pm 0.7$ |
| OrbitGrasp ($l = 3, m = 2$) | 14M | $\mathbf{98.3 \pm 0.7}$ | $\mathbf{98.8 \pm 0.6}$ | $\mathbf{96.7 \pm 1.1}$ | $\mathbf{97.9 \pm 0.5}$ |

Table 3. Comparison results between non-equivariant and equivariant networks with varying degrees of SH.

using height-thresholded FPS to form the neighborhoods $B_1, \ldots, B_k$. From the orbit of each point, we evenly sample 36 grasp poses and evaluate them based on the grasp quality function from the network. After filtering out unreachable and low-quality poses (i.e. grasp poses with a quality $<$ 0.95), we select the pose with the highest $Z$ value.

**Results:** We baseline our method against VNEdgeGrasp [8]. The results are shown in Table 2. In the single view *Packed* task, the GSR of our model outperformed VNEdgeGrasp by an average of 3.6% and by 5% in the single view *Pile* task. In the multi-view setting, our model's GSR outperformed by 5.1% on the *Packed* task and by 2.2% on the *Pile* task. We analyze the failure modes we encountered in Appendix F.

### 5.3 Ablation Study

In our ablation study, we measure the significance of network equivariance by comparing to Point-Net++ [48], a well-known non-equivariant model under a single random camera setting, keeping all settings identical except for the network structure. For fairness, we set up PointNet++ so that it has the same number of parameters as our method. We also measure the influence of different degrees of spherical harmonics, where higher degrees correspond to more basis functions and larger numbers of Fourier coefficients. The results, shown in Table 3, indicate that PointNet++ performs poorly (GSR of 60.5% versus 96.7% for the *Pile* task), presumably due to the lack of equivariance in SO(3) space. Interestingly, even with spherical harmonic basis functions of degree 1, our network performs well (GSR of 93.9% for the *Pile* task), a result we attribute to the capabilities of our UNet-based EquiFormerV2, which effectively captures both local and global information. As the degree increases, the performance improvement from additional parameters diminishes. Although the performance of ours ($l = 3, m = 2$) is slightly better (GSR of 96.7% for the *Pile* task) than ($l = 2, m = 2$), it requires nearly double the parameters. This trade-off should be considered when computing power is limited. We also analyze the impact of larger neighborhoods $B_i$ on performance in simulation, along with the effect of different training data generation methods (Mask versus FPS) in the real world, as detailed in Appendix G.

## 6 Conclusion and Limitations

In this paper, we propose *OrbitGrasp*, a grasp detection method that exploits SE(3)-equivariance to achieve state-of-the-art performance. Our model learns a *continuous* grasp function over $S^2$ for each point in the point cloud. Using a geometric prior, we constrain potential grasp poses around the surface normal direction of each candidate contact point, forming an orbit. By evaluating densely sampled poses along the orbit with the learned grasp function, multiple good grasp poses can be identified. Simulation experiments demonstrate that our model outperforms several strong baselines across different grasping tasks and settings. Physical experiments show high grasp success rates and good generalization across diverse objects.

Our work has several limitations. First, as discussed in Appendix E, the inference time is relatively long because PyTorch's lack of optimization for the geometric operations used in EquiFormerV2 and the need to evaluate multiple point neighborhoods, $B_1, \ldots, B_k$. One potential approach is to incorporate gauge equivariance [49] and predict directly in the tangent space [50], which would lower the computational complexity and dimensionality of the problem and speed up inference. Another limitation is the absence of a direct mechanism for constraining grasps to specific objects or object parts. Future work may address this challenge by integrating methods for conditioning on language or object crops.

## Acknowledgments

This work is supported in part by NSF 1750649, NSF 2107256, NSF 2134178, NSF 2314182, NSF 2409351, and NASA 80NSSC19K1474. Dian Wang is supported in part by the JPMorgan Chase PhD fellowship. The authors would like to thank Owen Howell and David Klee for their valuable discussions and advice on designing the $\mathrm{SE}(3)$-equivariant network and preparing the paper manuscript.

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

# A Mask-Based Sample Generation

To obtain object masks on a point cloud, all RGB-D images are first cropped to focus on the workspace area. The Segment Anything Model (SAM) is then applied to each RGB image to acquire 2D segmentation masks of all objects. Given the pixel-wise correspondence between the RGB image, masks, and depth map, we can map these 2D masks onto the point cloud. Note that each point $\mathbf{p}_i$ may belong to multiple masks, such as $\mathbf{p}_i \in \mathbf{M}_a$ and $\mathbf{p}_i \in \mathbf{M}_b$, where $\mathbf{M}_a$ and $\mathbf{M}_b$ are different masks. By combining the point clouds and mask information from multiple cameras, a raw mask-centric point cloud representation is reconstructed. After preprocessing and filtering, we finally obtain the refined mask-centric point cloud scene, illustrated in Figure 5.

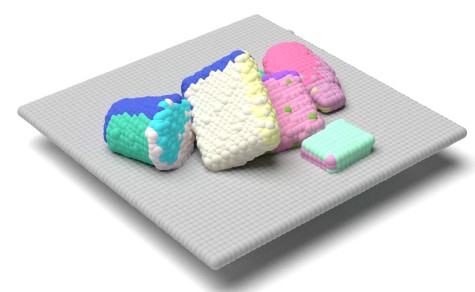

Figure 5. The mask-centric point cloud representation. Each mask is rendered in a distinct color. Points that belong to multiple masks are rendered with only one color.

In our implementation, we use the centers of the individual object masks as center points to construct $B_i$ for generating grasp poses during training data collection. This method provides more balanced object-level point sampling compared to the FPS-based method, as it assigns a center point to each object (mask), regardless of size. In contrast, FPS tends to allocate more points to larger objects and fewer to smaller ones, which leads to an uneven distribution of grasp points. Additionally, it avoids generating many unstable poses on the edges and corners of objects. Although FPS is still used to construct $B_i$ during inference, we found that our model effectively reduces the probability of grasping the edges or corners of objects.

# B Model Architecture

The detailed architecture of EquiFormerV2, as mentioned in *Implementing $\bar{\Gamma}$ as an Equivariant Neural Network* in Section 4, is illustrated in Figure 6. Compared to the original structure, we have made several key modifications. The original SO(3) embedding has been replaced with a single SO(3)-equivariant linear layer that directly takes the point coordinates and normal directions as input. This embedding is then used in subsequent blocks. After the first equivariant graph attention block, we apply FPS to downsample the point cloud. For each downsampled point, we use KNN to find its neighbors and build edges between them. During upsampling, we reverse these edges from each downsampling block by swapping the source and destination of these edges. This allows us to gradually transfer information from the downsampled points to those points in the upsampling blocks. Meanwhile, this process aggregates information from distant points and gradually extends this information to local points. The theoretical foundation and mathematical proof of EquiFormerV2 can be found in [17, 37, 51].

Figure 7 visualizes the process of evaluating grasp poses along the *orbit* of point $p$ in the point cloud. Starting with the point's normal vector $n_p$ and the set of Fourier Coefficients $\{\mathcal{F}_{l,p}^m\}$ output by the network, the spherical harmonics basis functions are multiplied with these coefficients to reconstruct the spherical signal on $S^2$, according to Equation 1. This signal serves as the grasp quality function, i.e., $f_p : S^2 \rightarrow \mathbb{R}$. The grasp sampler then uses the normal vector $n_p$ to generate a set of approach vectors $\{r_3 \in \overline{O}p = r_3 \in S^1 : n_p^\top r_3 = 0\}$, representing possible grasp poses. These vectors are then used to query $f_p$ to yield the quality of each grasp pose.

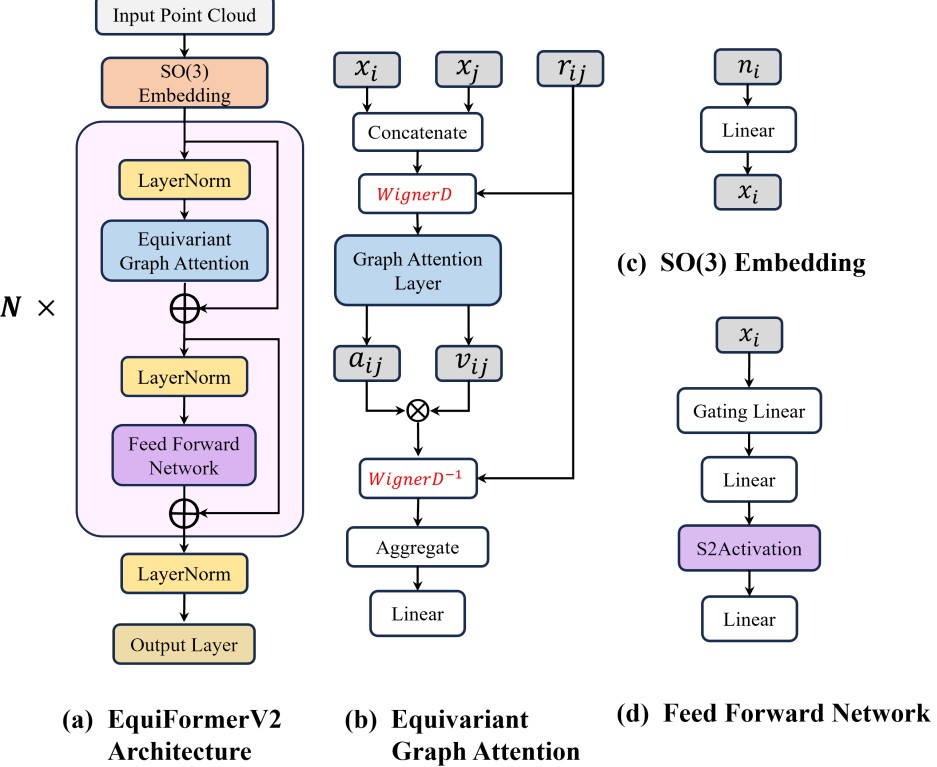

**(a) EquiFormerV2 Architecture**  **(b) Equivariant Graph Attention**  **(d) Feed Forward Network**

**(c) SO(3) Embedding**

Figure 6. **Overview of the EquiFormerV2 architecture.** (a) shows the overall structure of EquiFormerV2, while (b), (c), and (d) illustrate the submodules of (a). Multiple EquiFormerV2 blocks, incorporating FPS and KNN layers for connectivity, are stacked to form our UNet-style architecture.

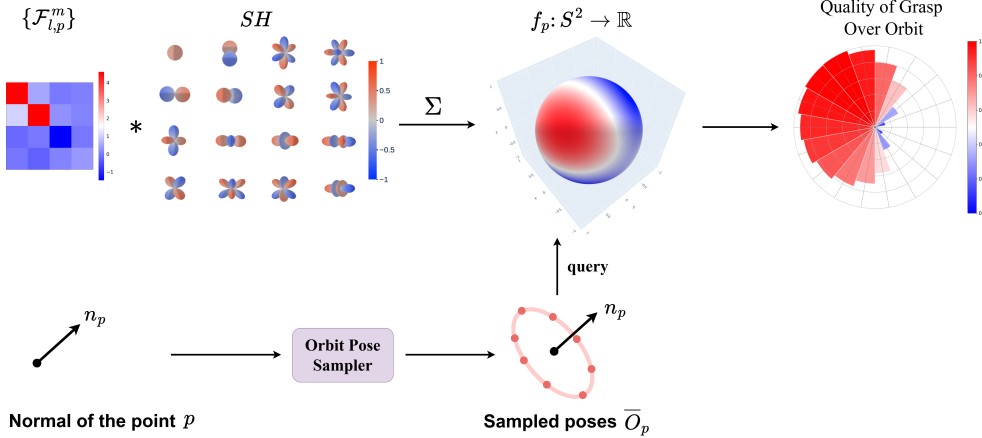

Figure 7. **Visualization of grasp poses sampling and evaluation.** To enhance clarity, this procedure is illustrated with a single point $p$ in the point cloud. The Fourier coefficients of $p$, denoted as $\{\mathcal{F}_{l,p}^m\}$, and the spherical harmonics basis functions $SH$, are used to reconstruct the grasp quality function $f_p\colon S^2 \to \mathbb{R}$ on the 2-sphere $S^2$, as described in Equation 1. The Orbit Pose Sampler generates potential grasp poses, which are then evaluated using $f_p(\cdot)$. The circle at the top right, viewed along the normal direction of point $p$, shows the grasp quality of each sampled pose, with redder colors indicating higher quality.

## C  Additional Details in Inferring Grasp Pose

The spherical signal $f_p$, output by the model, evaluates grasp orientations via the approach vector $r_3$. Notice that the unit vector $r_3$ is a direction that can be described by spherical coordinates $(\theta, \phi)$. While $f_p$ may hypothetically be evaluated at any $u \in S^1$, in practice, we evaluate it only at $u \in \overline{O}_p$ during training and inference. Using Equation 1, $f_p$ is evaluated over $\overline{O}_p$ and the pose that maximizes $f_p$ is selected through dense sampling of $\overline{O}_p$. To prevent the object from rotating when one finger makes contact before the other, we apply an offset along the gripper's closing direction to center the object between the fingers.

## D  Simulation Additional Details

We provide several figures (Figure 8, 9) to give more information about the simulation environment and grasp pose evaluation process. Figure 8(a) shows a pile scene with objects randomly dropped onto the table while Figure 8(b) indicates a packed scene with objects placed upright in random poses. Figure 9(a)(b)(c) presents the downsampled point cloud, the orbital sampled grasp poses, and the optimal pose within the entire scene, respectively.

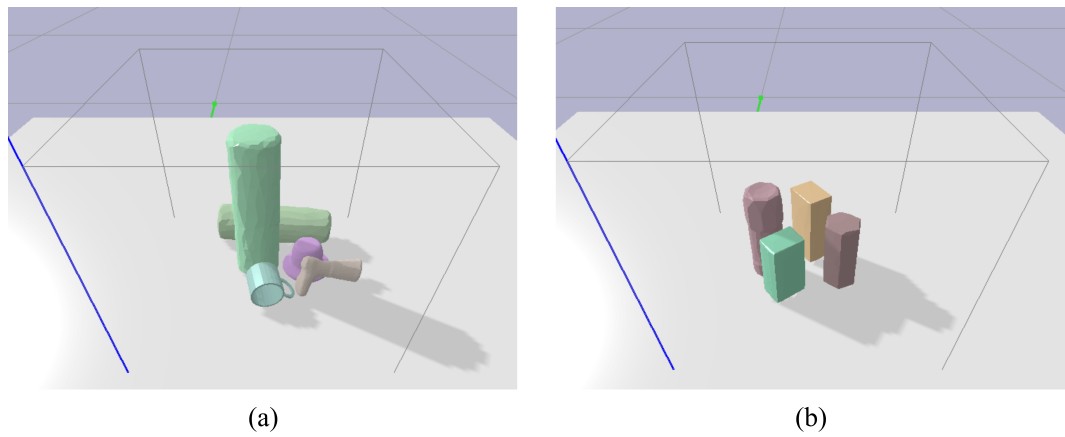

(a)                                                                      (b)

Figure 8. (a) and (b) illustrate examples of "pile" and "packed" scenes, respectively.

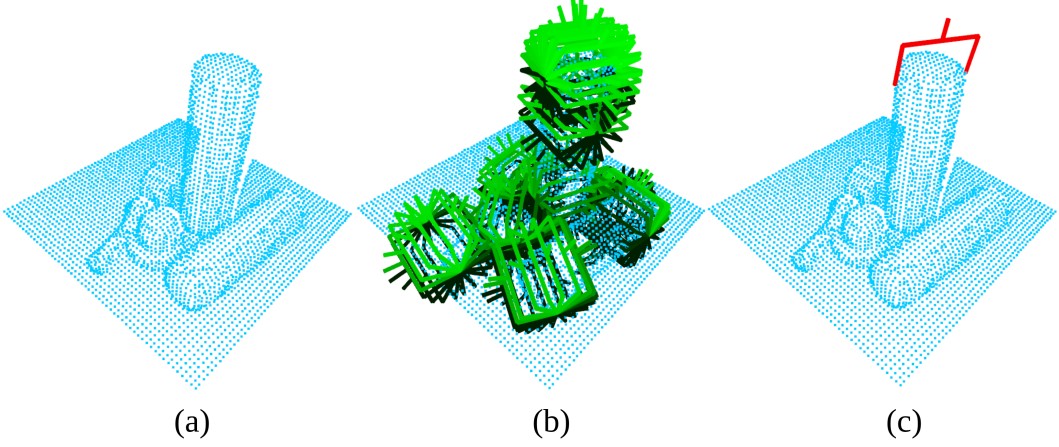

(a)                                     (b)                                     (c)

Figure 9. (a) The downsampled point cloud. (b) All sampled grasp poses at the points, with the pose that has the highest grasp quality in each of the 10 $\mathcal{B}_i$, where a more intense green color indicates higher quality. For simplicity, only 18 of the 36 sampled poses per point are displayed. (c) The best grasp poses selected from all $\mathcal{B}_{1,...,n}$.

| Setting | Data | Model | Packed | | Pile | |
| --- | --- | --- | --- | --- | --- | --- |
| | | | GSR (%) | DR (%) | GSR (%) | DR (%) |
| Single-View | 6M | $l=1,m=1$ | $94.2 \pm 1.2$ | $96.3 \pm 1.3$ | $93.9 \pm 0.5$ | $97.0 \pm 0.7$ |
| | | $l=2,m=2$ | $97.4 \pm 0.5$ | $98.5 \pm 0.6$ | $96.3 \pm 1.0$ | $97.6 \pm 0.7$ |
| | | $l=3,m=2$ | $98.3 \pm 0.7$ | $98.8 \pm 0.6$ | $96.7 \pm 1.1$ | $97.9 \pm 0.5$ |
| | 3M | $l=1,m=1$ | $95.2 \pm 0.9$ | $96.7 \pm 1.1$ | $94.7 \pm 1.0$ | $97.1 \pm 0.4$ |
| | | $l=2,m=2$ | $98.1 \pm 0.6$ | $98.7 \pm 0.5$ | $96.6 \pm 0.5$ | $98.0 \pm 0.5$ |
| | | $l=3,m=2$ | $98.4 \pm 0.5$ | $98.8 \pm 0.3$ | $96.3 \pm 0.3$ | $97.7 \pm 0.7$ |
| | 1M | $l=1,m=1$ | $94.7 \pm 1.2$ | $96.0 \pm 0.8$ | $94.2 \pm 0.6$ | $96.6 \pm 0.7$ |
| | | $l=2,m=2$ | $96.9 \pm 0.5$ | $98.3 \pm 0.8$ | $93.8 \pm 1.4$ | $96.2 \pm 1.2$ |
| | | $l=3,m=2$ | $97.3 \pm 0.7$ | $98.7 \pm 0.4$ | $94.1 \pm 0.6$ | $97.0 \pm 0.3$ |
| Multi-View | 6M | $l=1,m=1$ | $98.3 \pm 0.7$ | $98.4 \pm 0.6$ | $97.7 \pm 0.8$ | $97.9 \pm 0.7$ |
| | | $l=2,m=2$ | $99.1 \pm 0.4$ | $99.0 \pm 0.6$ | $98.2 \pm 0.4$ | $98.0 \pm 0.5$ |
| | | $l=3,m=2$ | $99.0 \pm 0.6$ | $99.2 \pm 0.3$ | $98.5 \pm 0.6$ | $98.2 \pm 0.6$ |
| | 3M | $l=1,m=1$ | $98.4 \pm 0.6$ | $98.6 \pm 0.2$ | $97.8 \pm 0.5$ | $97.5 \pm 0.8$ |
| | | $l=2,m=2$ | $98.6 \pm 0.6$ | $99.2 \pm 0.3$ | $97.9 \pm 0.5$ | $97.9 \pm 1.0$ |
| | | $l=3,m=2$ | $98.6 \pm 0.3$ | $99.1 \pm 0.5$ | $98.6 \pm 0.7$ | $98.5 \pm 0.5$ |
| | 1M | $l=1,m=1$ | $97.7 \pm 0.7$ | $97.9 \pm 0.8$ | $97.3 \pm 0.3$ | $97.8 \pm 0.6$ |
| | | $l=2,m=2$ | $98.7 \pm 0.3$ | $99.1 \pm 0.4$ | $98.2 \pm 0.4$ | $98.2 \pm 0.6$ |
| | | $l=3,m=2$ | $99.0 \pm 0.7$ | $99.2 \pm 0.4$ | $98.4 \pm 0.2$ | $98.0 \pm 0.4$ |

Table 4. Comparison of results across different amounts of training data, camera settings, and model SH degrees.

## E  Efficiency and Inference Time Analysis

**Training Data Efficiency.**  While we use 6M labeled grasp poses (both positive and negative) to train our model to ensure sufficient data for learning and to prevent data limitations from being a bottleneck, our method also performs well with less training data. To demonstrate this, we trained our model with 3M data points (matching the amount used in EdgeGrasp [8] and ICGNet [9]) and with 1M data points (a significantly smaller dataset). Table 4 indicates that our model consistently outperforms the baselines across all scenarios. Even with just 1M data points, our method still performs strongly and surpasses other methods, as seen in Table 1. These findings emphasize the efficiency and robustness of our method.

**Inference Time.**  Rapid inference of grasping poses is critical for grasp detection methods. Although our approach requires dense sampling of grasp poses for each point in the point cloud, this process is efficient because it only requires a single forward pass through the model per region $\mathcal{B}_i$. Specifically, one forward pass yields a vector of spherical harmonic coefficients at each point in $\mathcal{B}_i$. To find the optimal grasp, we identify the argmax of the grasp quality function by sampling values on $S^2$ and taking the maximum. This sampling is performed efficiently through matrix multiplication with spherical basis functions, as described in Equation 1, which eliminates the need for additional forward passes.

To validate our approach efficiency, we report the inference, pre-processing, and post-processing times under a single-view setting in Table 5. For $k = 1$, object-centric grasping is employed, where a segmentation model is used to locate the target object and crop the point cloud for prediction. This is similar to conditional grasping, as it focuses on a specific target. For $k = 10$, 10 regions $\mathcal{B}_{1,...,10}$ are generated. This configuration helps our method to understand the entire scene and generate potential grasp poses for all points, which is suitable for scenarios with multiple objects or when the target is not predefined. The quantitative results suggest that our method can respond within 0.25 seconds when grasping a target object and evaluate grasp poses for nearly the entire scene in 1.5 seconds—fast enough for typical open-loop grasping tasks. While our inference time is comparable to VN-Edgegrasp for $k = 1$, it is slower for $k = 10$. However, given that our method outperforms theirs in accuracy and can evaluate almost the entire scene, this difference in speed is acceptable.

| Model | number of k | Inference | Latency |
|---|---|---|---|
| $l = 3, m = 2$ | $k = 1$ | 0.126 s | 0.133 s |
| | $k = 10$ | 0.856 s | 0.678 s |
| $l = 2, m = 2$ | $k = 1$ | 0.122 s | 0.116 s |
| | $k = 10$ | 0.743 s | 0.659 s |
| $l = 1, m = 1$ | $k = 1$ | 0.095 s | 0.117 s |
| | $k = 10$ | 0.455 s | 0.654 s |
| VN-EdgeGrasp [8] | ✗ | 0.153 s | 0.154 s |

Table 5. The runtime for different degrees of SH and the baseline is measured on an RTX 4090 GPU with an Intel i9-12900k CPU. 'Inference' refers to the model's running time, while 'Latency' includes both pre-processing and post-processing times.

## F   Physical Experiments Additional Details

**Implementation Details.**   Although our design initially selects the grasp pose with the highest $Z$ value after filtering, we observed that this highest $Z$ pose can sometimes be unstable on the physical robot, unlike in simulations. To address this, since our method generates a series of grasp poses for each point, we also consider poses within a 3cm range below the highest one. If a pose within this range offers the best grasp quality (i.e., surpasses the highest pose and all other candidates), we select it instead. This approach effectively reduces failures caused by weak grasps and mitigates the sim-to-real gap.

**Failure Mode Analysis.**   In the single-view setting, we observe that the GSR of our method for the *Packed* scene is lower than that for the *Pile* scene, which is inconsistent with the simulation results. The primary failure mode ($5/8$) involves a white bottle that blends with the table mat color. This blending results in inaccurate depth and point cloud shape estimation by the camera, which prevents the gripper from being inserted deeply enough to provide sufficient friction. For the *Pile* task, the primary reason for failure is the thickness of the objects. Thin objects tend to merge with the table in the point cloud, distorting the shape of objects and leading to inaccurate normal estimation and unreasonable poses generated by our model. Additionally, smooth surfaces and specific shapes, like those of stones, lead to insufficient friction and cause them to slip from the gripper. In the multi-view setting, the overall tendency of failure is similar to the single-view setting. While multiple perspectives help reduce distortion, introducing more cameras also introduces calibration errors. These errors, in turn, transfer to noises that appear in the point cloud.

## G   Full Ablation Results

**Larger Point Cloud as Input to Provide Sufficient Context.**   As mentioned in *Sampling k Center Points* in Section 4, using a larger point cloud $B_i = \mathcal{N}(c_i, m)$ instead of just the local point cloud $\mathcal{B}_i = \mathcal{B}(c_i, r_l)$ is crucial for eliminating boundary effects by providing more context for evaluating the grasp quality of each point in $\mathcal{B}_i$. To evaluate this, we compared the performance of these two input formats. The results, shown in Figure 10, indicate that using the larger point cloud as input significantly improves performance, with a 40% reduction in validation loss and a 10% increase in prediction accuracy. These findings highlight the importance of larger point clouds for providing sufficient context.

**Performance Comparison Between Mask-Based and FPS-Based Training Data.**   We compared the effects of mask-based versus FPS-based training data. Although both methods perform similarly in simulations, differences appear in real-world experiments. As shown in Figure 11, with the same input, the grasp quality distribution from the mask-based trained network is more uniform and centered around the object's center of mass (e.g., the banana, hammer, and shoe). This indicates that the mask-based trained network can evaluate a wide range of grasp poses more effectively. In contrast,

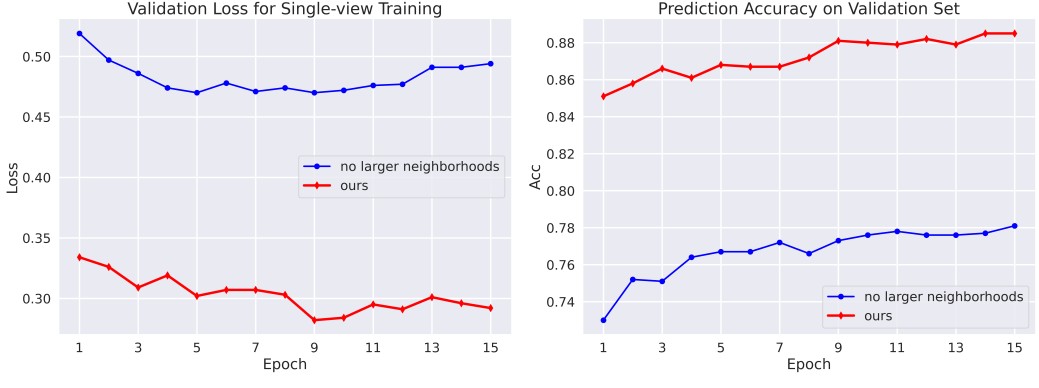

(a). The validation loss trends.

(b). The predicted accuracy of all validation poses.

Figure 10. **Ablation Study Results with Larger Point Cloud Input.** The prediction accuracy on the validation set is calculated by evaluating all grasp poses, including both positive and negative poses, to determine the overall accuracy.

the FPS-based trained network tends to produce grasps biased toward the object's edges or specific small regions. These edge-focused poses are generally less stable than those around the center of mass. We interpret this as a result of FPS-based training lacking object-centric awareness, which causes the network to focus on specific areas and struggles to assess grasps across the object. Mask-based training data, however, incorporates object-centric information. This enables the network to evaluate grasp poses more evenly across the entire object. Therefore, even when FPS is used for input during inference, the network trained with mask-based data maintains enough robustness to handle unseen geometric variations.

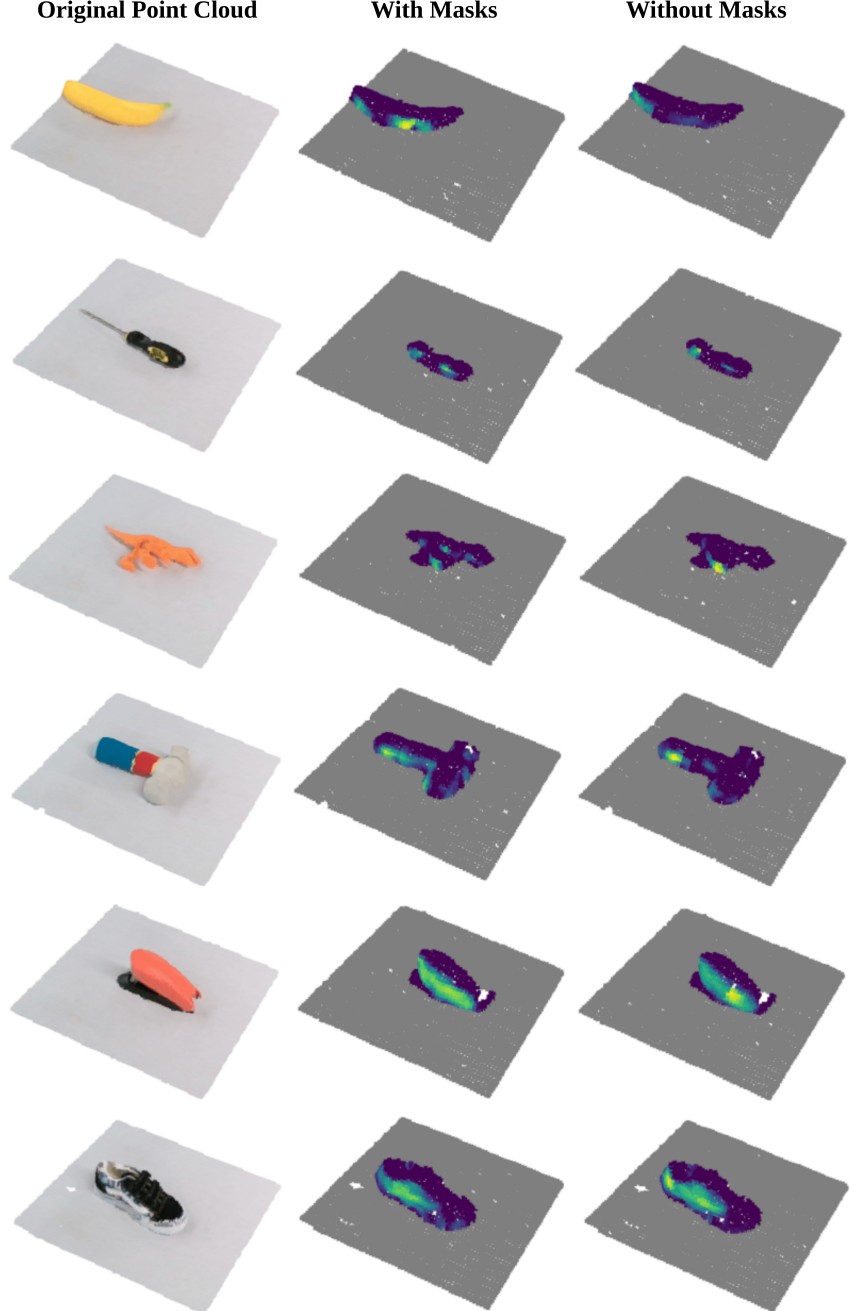

Figure 11. **Grasp quality distribution for different training data generation strategies.** The highest grasp quality of all sampled poses at each point represents that point's quality. The left column displays the original point cloud. The middle column shows predictions from the network trained with mask-based data. The right column shows predictions from the network trained with FPS-based data.

