# OpenReview forum: "OrbitGrasp: SE(3)-Equivariant Grasp Learning"
_robot-learning.org/CoRL/2024/Conference — CoRL 2024_

### Official Review · Reviewer_Zbw2 · 2024-06-26

**Originality:** 3
**Technical Quality:** 1
**Clarity Of Presentation:** 2
**Potential Impact:** 3
**Recommendation:** 3
**Confidence:** 4

**Review:**

# Strengths
The paper is clearly written, especially regarding the mathematical formalism and notation.

The experimental analysis is comprehensive. It includes experiments in both simulated and real-world environments, with comparisons to several baselines.

# Weaknesses
I believe that a major weakness of the proposed approach is the constraint of sampling grasp poses where the y-axis of the hand (green axis in Figure 3) is parallel to the surface normal. This is restrictive, and not realistic for several objects. For example, how does that allow to grasp a cup by its handle?

The above limitation is also reflected by a redundancy in the proposed model. Specifically, as explained in Section 4.6, the model outputs a grasp pose evaluation function $f_p: S^2 \rightarrow [0,1]$ for each $p$ in the point-cloud. However, because of the above constraint, only a one-dimensional submanifold of the sphere is relevant – specifically, a copy of $S^1$ in $S^2$. This makes the model’s output redundant. Why not designing a model that outputs a function defined directly on $S^1$ instead?

Another weakness is the fact that $f_p$ is not normalized, even though it represents a grasp quality. Since grasp quality is, intuitively, a distribution from which one wants to sample, I would expect some form of normalization. Even further, I think that the lack of normalization is reflected by an issue in the training loss. Specifically, as explained in Section 5 (Training Details), the model is trained via binary-cross entropy on each output of $f_p$ independently. In other words, the values $f_p(x) \in [0,1]$, $x \in S^2$, are interpreted as independent Bernoulli distributions that detect a grasp. But I believe this is a mathematically ill-posed optimization problem, and that some dependency as $x$ varies is necessary (for example, normalizing $f_p$ to integrate to $1$ over $S^2$). Since the model is trained only on a dataset of ‘positive’ grasps (i.e.,  for which ideally $f_p$ outputs $1$), why doesn’t the constant function $f_p(x) = 1$ minimize the loss?



# Additional Comments
I believe that the paragraph in the background section on the representation theory of $SO(3)$ (lines 96-104) is redundant. This paragraph introduces plenty of mathematical notation – especially for Wigner matrices. I understand, as explained in the next paragraph, that these matrices are used to formalize the equivariance properties of spherical harmonics, but they are not used anywhere else in the paper. Therefore, I think the amount of formalism introduced is not justified by its use in the paper.

**Quality Of The Limitations Section:**

2

**Questions For Rebuttal:**

I would like the authors to address the questions above regarding 1) the constraint and redundancy in the model, and 2) the normalization issue, especially as related to the issue in the optimization problem.

**Robotics Focus:**

4

**Summary Of Paper:**

The work proposes an $SE(3)$-equivariant model predicting grasp qualities, given a point-cloud representing an object. The grasp quality is modelled as a function over the sphere, decomposed in spherical harmonics.

**Summary Of Recommendation:**

I believe that the approach, despite interesting and original, suffers from the technical issues explained above. Therefore, I lean towards rejecting it in the current state.

---

### Official Review · Reviewer_k5Je · 2024-06-27
**Interesting approach to learning 6DoF grasping.**

**Originality:** 4
**Technical Quality:** 4
**Clarity Of Presentation:** 4
**Potential Impact:** 3
**Recommendation:** 3
**Confidence:** 4

**Review:**

Strength:
1. A novel equivariance approach to full 6DoF grasping, with a sufficient demonstration of the method on real robot environments in a tabletop setting.
2. The reviewer thinks that representing point-level grasp quality function via spherical harmonics is an interesting idea.
3. Sufficient demonstration of model performance and comparison with baseline methods.

Weakness:
1. The proposed method needs to split a large point cloud into smaller sub-point clouds, which may not be as efficient compared to other methods that do not rely on such equivariance.
2. All experiments are done in a tabletop setting, which in many cases does not necessarily justify the use of 6DoF grasping. A comparison with prior SE(2) methods [1] on grasp success and/or simulation of moving objects under more complex motion constraints (i.e. shelves) would further justify the improvement of the proposed method.

[1] X. Zhu, D. Wang, O. Biza, G. Su, R. Walters, and R. Platt. Sample efficient grasp learning using equivariant models. arXiv preprint arXiv:2202.09468, 2022.

**Quality Of The Limitations Section:**

3

**Questions For Rebuttal:**

1. What are some challenges you run into when performing sim2real transfer? How large is each $B_i$? Does increasing or decreasing the number of sub-point clouds change the performance?
2. Given that grasps are sampled on the point cloud (object surface) instead of inside the object, will this parametrization result in more collision and object rotation when closing the gripper?
3. In the simulation, all the $B_i$ are generated via SAM: “In our experiments, we generated the center points using object masks for training and FPS for inference.” (line 155-156). Why isn’t SAM run in real?

**Robotics Focus:**

4

**Summary Of Paper:**

The paper develops an SE(3) equivariant grasp synthesis network that takes in point cloud input and generates full 6DoF grasps. The approach learns a mapping from each point in the point cloud to a grasp quality function on the sphere centering at the point, where the function is encoded via spherical harmonics. At inference time, the point cloud is separated into small regions of point clouds, and each is inference separately. The authors compare the results against another SE(3) invariant

**Summary Of Recommendation:**

While there are some limitations, such as the need to split large point clouds and the focus on tabletop settings, these do not significantly reduce the paper's contributions. The work presents a new perspective on grasp synthesis and shows promise for learning better grasp representations. Thus I recommend accepting the paper.

---

### Official Review · Reviewer_RMP5 · 2024-07-20
**Major concern over the efficiency of sampling-based training and inference**

**Originality:** 3
**Technical Quality:** 2
**Clarity Of Presentation:** 3
**Potential Impact:** 2
**Recommendation:** 3
**Confidence:** 4

**Review:**

The content of the paper is organized well. The presentation of the problem and the method is mostly clear. The literature review and background are proper and help readers understand the context.

Methodology-wise, my major concern is efficiency. While the Fourier coefficient output models a continuous function, the pooling of this function relies on sampling. It makes the evaluation of the learned model more tedious. The grasping quality of a single point needs multiple evaluations of the sampled approaching orientations, and it has to be done for all points to find the optimal grasping pose. This is a very inefficient process, both for training and inference. In order to train the network, this paper generated 6M grasp poses for each camera setting, which is a lot for the grasping-learning task. In my understanding, the superior performance in the tables is likely due to the dense sampling of the grasping points and orientations, which could significantly sacrifice efficiency. The authors should include the actual inference time for accomplishing the task, in comparison with the baselines.

On the other hand, the heuristics used in sampling the grasping pose could be a limiting factor. The hand y-axis Is set to the surface normal direction at the grasping point. While it could alleviate the efficiency problem, it may sacrifice the flexibility of grasping. The surface normal of a point could be unreliable due to observation noise or specific local geometry. Could you comment and analyze on this limitation?

**Quality Of The Limitations Section:**

2

**Questions For Rebuttal:**

See above. Please answer the questions in terms of efficiency and the surface-normal heuristics.

**Robotics Focus:**

4

**Summary Of Paper:**

This paper proposed to learn the grasping quality through an SE(3)-equivariant model. Using fourier coefficients as the output, the learned grasping quality is a continuous function on the 2-sphere, from which the authors claim better generalization.

**Summary Of Recommendation:**

The limitation in efficiency could fundamentally limit the value of this work.

---

### Author Rebuttal · Authors · 2024-08-13

We have attached a zip file that includes the updated paper with the relevant changes highlighted.

---

### Decision · Program_Chairs · 2024-09-04

**Decision:**

Accept

**Comment:**

Thank you for your submission to CoRL 2024. The reviewers appreciated many aspects of the paper, including its clarity, use of formalisms, the novelty of the approach, method aspects (e.g., use of spherical harmonics), and the wealth of real-world and simulated demonstrations and comparisons to baselines.

Reviewers raised concerns about efficiency, heuristic and sampling limitations (including imposed constraints), and would like to see more experiments requiring full 6-DOF grasping.

However, the reviewers and ACs found the authors' thoughtful and thorough responses convincing. While there are lingering concerns (which we encourage the authors to address for the camera-ready version), several reviewers raised their final rating.